# Differential Strain-Specific Responses of *Trichoderma* spp. in Mycoparasitism, Chitinase Activity, and Volatiles Production Against *Moniliophthora* spp.

**DOI:** 10.3390/microorganisms13071499

**Published:** 2025-06-27

**Authors:** María F. Garcés-Moncayo, Christian A. Romero, Simón Pérez-Martínez, Carlos Noceda, Luís L. Galarza, Daynet Sosa del Castillo

**Affiliations:** 1Centro de Investigaciones Biotecnológicas del Ecuador (CIBE), Escuela Superior Politécnica del Litoral (ESPOL), Campus Gustavo Galindo, Km. 30.5 Vía Perimetral, Guayaquil P.O. Box 09-01-5863, Ecuador; mgarcesm2@unemi.edu.ec; 2Facultad de Ciencias e Ingeniería, Universidad Estatal de Milagro (UNEMI), Milagro 091050, Ecuador; 3Carrera de Enfermería, Universidad Bolivariana del Ecuador (UBE), Km 5.5 Vía Durán-Yaguachi, Durán 092405, Ecuador; caromerob@ube.edu.ec; 4Departamento de Ciencias de la Vida y de la Agricultura, Biología Celular y Molecular de Plantas (BIOCEMP)/Biotecnología Industrial, Universidad de las Fuerzas Armadas (ESPE), Av. General Rumiñahui s/n, Sangolquí, Santo Domingo 230153, Ecuador; cmnoceda@espe.edu.ec; 5Non-Institutional Competence Focus (NICFocus) ‘Functional Cell Reprogramming and Organism Plasticity’ (FunCROP), Coordinated from Foros de Vale de Figueira, 3400-641 Alentejo, Portugal; 6Facultad de Ciencias de la Vida (FCV), Escuela Superior Politécnica del Litoral (ESPOL), Campus Gustavo Galindo, Km. 30.5 Vía Perimetral, Guayaquil P.O. Box 09-01-5863, Ecuador; llgalarz@espol.edu.ec

**Keywords:** *Moniliophthora roreri*, *Moniliophthora perniciosa*, biocontrol, cacao pathogen, volatile organic compound (VOC), chitinase

## Abstract

*Moniliophthora roreri* (MR, frosty pod rot) and *M. perniciosa* (MP, witches’ broom disease) pose critical threats to cacao production in Latin America. This study explores the biocontrol potential of *Trichoderma* spp. strains against these pathogens through exploratory analysis of mycoparasitism, chitinolytic activity, and volatile organic compound (VOC) production. Dual-culture assays revealed species-specific antagonism, but C2A/C4B showed a dual-pathogen efficacy (>93% of *Monioliopthora* inhibition). Chitinase activity revealed C4A/C1 strains as exceptional producers (72 mg/mL NAGA vs. MR and 94 mg/mL vs. MP, respectively). GC-MS analysis identified pathogen-modulated VOC dynamics: hexadecanoic acid dominated in 80% *Trichoderma* solo-cultures (up to 26.65% peak area in C3B). MP showed 18.4-fold higher abundance of hexadecanoic acid than MR (0.23%). In 90% of dual-culture with MR and MP, HDA was detected as the most abundant. Functional specialization was evident. C4A and C1 prioritized chitinase production growing on MR and MP cell walls (respectively), whereas C9 excelled in antifungal hexadecanoic acid synthesis in confrontation with both pathogens. Complementary strengths among strains—enzymatic activity in C4A/C4B versus volatile-mediated inhibition in C9—suggest niche partitioning, supporting a consortium-based approach for robust biocontrol. This study provides preliminary evidence for the biocontrol potential of several *Trichoderma* strains, showing possible complementary modes of action.

## 1. Introduction

Cacao (*Theobroma cacao* L.) represents a crop of major economic importance across tropical regions. However, its productivity is severely compromised by fungal pathogens, particularly *Moniliophthora roreri* (MR), the causal agent of frosty pod rot, and *M. perniciosa* (MP), responsible for witches’ broom disease. These pathogens can cause devastating yield losses, exceeding 80% in key producing countries such as Ecuador, Brazil, and Costa Rica [1,2]. While synthetic fungicides remain the primary control strategy, their widespread use raises concerns about environmental sustainability, fungicide resistance, and increasing production costs, highlighting the need for effective biological alternatives [3].

Among potential biocontrol agents, *Trichoderma* species stand out due to their broad spectrum of antagonistic mechanisms [4]. Unlike fungi that depend primarily on hyphal interference (HI)—a contact-dependent and often transient mechanism with limited efficacy under field conditions [5]—*Trichoderma* employs a multifaceted strategy. These include (i) secretion of cell wall-degrading enzymes (CWDEs) such as chitinases and glucanases, (ii) production of inhibitory secondary metabolites and volatile organic compounds (VOCs), and (iii) induction of systemic resistance in host plants [6,7]. Notably, *Trichoderma* can parasitize and exploit pathogen biomass as a nutrient source, allowing sustained pathogen suppression beyond simple contact-based inhibition [4,8].

Chitin, although constituting only 10–20% of the filamentous fungal cell wall, is a critical structural polysaccharide conferring tensile strength and integrity [9]. Disruption of chitin synthesis leads to cell wall instability and increased susceptibility to environmental stress (malformation and osmotically unstable). Chitin and its derivatives are also potent elicitors of plant defense responses [10]. *Trichoderma* chitinases (EC 3.2.1.14) and β-1,3-glucanases (EC 3.2.1.39) are recognized as key enzymes in mycoparasitism, facilitating the breakdown of phytopathogenic fungal cell walls [11,12,13]. Cytochemical localization of N-acetylglucosamine residues released has confirmed rapid hydrolysis of wall-bound chitin by extracellular chitinases during *Trichoderma* attack [14].

Despite the proven efficacy of *Trichoderma* against various phytopathogens, its specific interactions with *Moniliophthora* spp. remain insufficiently understood. The effectiveness of *Trichoderma* chitinases against the unique chitin/protein matrices of *Moniliophthora* cell walls is not well understood and may require specialized enzymatic adaptations [15]. Additionally, the role of VOCs in *Trichoderma*–*Moniliophthora* interactions is largely unexplored, although such compounds are known to inhibit pathogen growth and trigger plant defenses [16]. For instance, VOCs produced by cacao-associated endophytic bacteria have shown strong inhibitory effects on MR growth and sporulation [17], and a metabolic exchange occurs during in vitro confrontation between MR and *T. harzianum*.

This study aims to analyze the antagonistic potential of ten *Trichoderma* strains against *M. roreri* and *M. perniciosa* by (i) assessing mycoparasitic activity in dual culture assays, (ii) quantifying chitinolytic enzyme activity using pathogen-derived chitin as substrate, and (iii) identifying VOCs produced during pathogen interaction using gas chromatography–mass spectrometry (GC-MS).

## 2. Materials and Methods

### 2.1. Fungal Strains and Culture Conditions

The study utilized ten *Trichoderma* strains alongside two pathogenic *Moniliophthora* species (MR and MP), all obtained from the Culture Collection of Microorganisms held at the Biotechnology Research Center of Ecuador (CCM-CIBE). These strains were originally isolated from the rhizosphere of cacao plants in Naranjal, Ecuador. Strain identification was confirmed through genus by sequencing of the internal transcribed spacer (ITS) region using primers ITS1 and ITS4 [18]. Nine out of the ten *Trichoderma* isolates exhibited high molecular similarity (99–100% sequence identity) to at least four known species: *T. spirale* (strains C3B, C5, C8, C9, C10), *T. harzianum* (C1, C4A), *T. reesei* (C2A), and *T. ghanense* (C4B). The remaining strain C3A did not match any recognized *Trichoderma* species with significant similarity. Due to the limited taxonomic resolution of the ITS region, all strains were conservatively classified at the genus level, with species assignments considered provisional. For long-term preservation, cultures were stored in 10% glycerol at −80 °C. Fungal strains were reactivated through three successive subcultures on 90 mm diameter Petri dishes containing potato dextrose agar (PDA, Difco Laboratories, Sparks, MD, USA) supplemented with 10 μg/mL gentamicin. Plates were incubated in the dark at 28 °C. The *Trichoderma* and *Moniliophthora* strains employed in this study have a history of prior use in both published and unpublished laboratory and field investigations [19,20,21,22].

### 2.2. Dual Culture Antagonism Assay

Antagonistic potential was evaluated using a dual-culture plate assay where 5 mm mycelial plugs of 7-day-old *Moniliophthora* cultures were inoculated on one side of 90 mm PDA plates [22]. After allowing 7 days for pathogen establishment at 28 °C, 4-day-old *Trichoderma* plugs were placed 6 cm opposite the pathogen. Control plates were prepared by inoculating mycelial plugs of MR, MP, and each *Trichoderma* strain at the edge of 90 mm diameter Petri dishes containing PDA, and incubated under the aforementioned conditions. Mycelial growth of both pathogenic and beneficial strains was measured on the day one of the colonies fully colonized the Petri dish surface [20]. Antagonism was quantified through percent of pathogen growth inhibition (PPI), calculated as PPI = 100 × [PC − (FP − EP)]/PC, where PC represents radial growth of the pathogen (control, without antagonist), FP the confronted pathogen radial growth, and EP the radius reached during the 7 days of establishment of the pathogen [18].

### 2.3. Chitinolytic Activity Analysis

#### 2.3.1. Chitin Preparation from Mycelium Cell Walls

To isolate chitin from MR and MP mycelium cell walls, 100 mL Erlenmeyer flasks, each containing 50 mL of potato dextrose broth (PDB, Difco Laboratories, Sparks, MD, USA), were inoculated with 8 mm mycelial discs of each pathogen. Following a 20-day incubation period at 28 °C and 120 rpm, mycelia were harvested via vacuum filtration using Whatman No. 1 filter paper (Cytiva, Elicrom, Ecuador), washed three times with sterile distilled water, pulverized in liquid nitrogen, and dried at 40 °C for 24 h [23]. Subsequently, colloidal chitin was prepared by hydrolyzing 5 g of mycelium overnight with 60 mL of concentrated HCl under constant agitation at 4 °C. The resulting mixture was treated with chilled (−20 °C) 95% ethanol and rapidly stirred for 5 min, followed by centrifugation at 3000 rpm for 20 min at 4 °C. The resulting colloidal chitin pellet was washed repeatedly with sterile distilled water until the ethanol odor was no longer detectable, with 5 min centrifugation steps at 3000 rpm and 4 °C after each wash. The final chitin product was stored at 4 °C until further use [15,24].

#### 2.3.2. Commercial Chitin Preparation from Shrimp Shells

Five grams of commercial chitin (HIMEDIA, Mumbai, India) derived from shrimp shells were hydrolyzed overnight with 60 mL of concentrated HCl under constant agitation at 4 °C to break down the β-(1→4) glycosidic bonds, decrystallize chitin, increase its solubility, and accelerate subsequent enzyme hydrolysis. The resulting mixture was treated with chilled (–20 °C) 95% ethanol to precipitate the colloidal chitin out of the solution and rapidly stirred for 5 min. The chitin suspension was treated as described before in Section 2.3.1.

#### 2.3.3. Semi-Quantitative Determination of *Trichoderma* spp. Chitinolytic Activity

Chitinase detection medium was prepared with a basal solution (0.3 g MgSO_4_.7H_2_O, 3.0 g (NH_4_)_2_SO_4_, 2.0 g KH_2_PO_4_, 1.0 g monohydrated citric acid, 15 g agar, 200 µL Tween-20, 4.5 g, 1 L) and 1 L of distilled water, supplemented with 4.5 g of chitin (either commercial shrimp or derived from MR and MP cell walls) and 0.15 g bromocresol purple dye. The pH of the medium was adjusted to 4.7 prior to sterilization at 121 °C for 15 min. Chitinolytic activity was assessed by placing fresh *Trichoderma* culture plugs at the center of Petri dishes containing 25 mL of the chitinase detection medium. Plates were incubated at 28 °C for 3 days, and the formation of a color change zone was recorded every 24 h [15].

#### 2.3.4. Quantification of Total *Trichoderma* spp. Chitinase Activity

To induce chitinolytic activity in *Trichoderma* strains, 4.5 g of colloidal chitin (either commercial or MR- or MP-derived cell walls) was used as the sole carbon source. Eight-millimeter mycelial plugs from 4-day-old *Trichoderma* cultures were individually inoculated into 50 mL flasks containing a basal medium without bromocresol purple dye. Following incubation at 28 °C and 200 rpm for 5 days, the cultures were filtered through 0.22 μm membrane filters and stored at −20 °C until further analysis. Total chitinase activity was determined as follows: 1 mL of filtered culture supernatant was combined with 0.3 mL of 1 M sodium acetate buffer (pH 4.6) and 0.2 mL of colloidal chitin in 50 mL flasks. The mixture was incubated at 40 °C for 20 h and then centrifuged at 13,000 rpm for 5 min at 6 °C. Subsequently, 0.75 mL of the resulting supernatant, 0.25 mL of 1% dinitrosalicylic acid in 0.7 M NaOH, and 0.1 mL of 10 M NaOH were mixed in 1.5 mL microcentrifuge tubes and heated at 100 °C for 5 min in a water bath. After cooling to room temperature, the absorbance was measured at 585 nm wavelength. A standard curve generated using N-acetyl-β-D-glucosamine (NAGA) was used to quantify the reducing saccharide concentration. Chitinase activity was expressed as mg/mL of released NAGA [15,25].

### 2.4. Volatile Organic Compounds Analysis

#### 2.4.1. Volatiles Extraction

Extractions from the confrontation zones *Trichoderma*-MR/MP using a modified protocol were performed [26]. Samples for volatile (VOCs) extraction were taken from the plates used in Section 2.2 and from the solo-culture of each strain. *Trichoderma* spp. and *Moniliophthora* spp. were cultivated in solo-culture for 7 and 15 days, respectively. This differential cultivation period was necessary due to the slower growth rate of the *Moniliophthora* pathogens [22]. Solid medium containing fungal mycelia was aseptically excised into approximately 10 × 10 mm fragments. These fragments were transferred to 250 mL Erlenmeyer flasks and submerged in 30 mL of ethyl acetate, and constant agitation (120 rpm/48 h). Subsequently, the ethyl acetate extracts were concentrated to 2 mL via rotary evaporation under reduced pressure at 40 °C and then filtered through 0.22 μm membrane filters [20,26,27]. Control extracts were prepared using the same protocol, but with uninoculated PDA culture media. All extracts were stored at 4 °C until GC-MS analysis.

#### 2.4.2. VOCs Analysis

To explore the secondary metabolites produced by *Trichoderma* spp. that may contribute to their antagonistic activity against *Moniliophthora* pathogens, ethyl acetate extracts were analyzed using gas chromatography-mass spectrometry (GC-MS) with a non-polar column. Compounds with a mass spectral match quality score exceeding 90% were selected for downstream analysis, ensuring high confidence in compound identification. The chromatograms revealed an average of 38 compounds in each extract. However, subsequent analyses focused only on molecules with antifungal or antimicrobial activity for potential biocontrol.

VOCs were separated and identified using an Agilent Technologies 7890A GC (Santa Clara, CA, USA) coupled to a 5975C triple-axis mass spectrometer, following a modified method [20]. Separation was achieved on a DB-5MS non-polar capillary column (30 m × 0.25 mm × 0.25 μm) with phenyl dimethyl polysiloxane stationary phase (0.25 μm film thickness). The operating conditions were as follows: ultra-high purity helium carrier gas at 1.2 mL/min; initial oven temperature 70 °C (2 min hold), ramped at 5 °C/min to 285 °C, and held at 250 °C for 5 min; injector temperature 260 °C (splitless mode). The MS transfer line and ion source temperatures were maintained at 305 °C and 230 °C, respectively, with an electron ionization energy of 70 eV. Mass spectra were acquired in full scan mode over a mass range of 40–700 m/z (mass/charge) using a quadrupole mass analyzer. Compound identification was performed by comparing mass spectra to reference spectra from the Wiley 9 and NIST 2011 libraries. Analysis was conducted to compare samples and determine the dominant compounds in substrate extracts from *Trichoderma* spp. and *Moniliophthora* spp. monocultures and co-cultures. The peak area (%) represents a quantitative proportion of the predicted compound to the total of crude extracts.

### 2.5. Statistical Analysis

Given the exploratory nature of this study and constraints in biological material availability, we analyzed single-replicate datasets from ten *Trichoderma* strains using an exploratory framework appropriate for initial screening purposes. While this precludes formal inferential statistics, data were analyzed using descriptive statistics focused on range, means, and patterns across strains to identify general trends.

To visualize differences in volatile organic compound (VOC) profiles across strains during pathogen confrontation, hierarchical clustering analysis was performed. Bray–Curtis dissimilarity was selected as the distance metric due to its robustness in handling sparse, zero-inflated ecological data sets, such as VOC profiles. Cluster formation employed UPGMA (Unweighted Pair Group Method with Arithmetic Mean) linkage using the software PAST (v5.2.1, 2025).

In addition, Principal Component Analysis (PCA) was conducted using InfoStat (v2020) to reduce dimensionality and uncover underlying patterns across biocontrol-related traits, including colony antagonism, chitinase activity, and volatile production. These multivariate analyses allowed us to group strains by shared phenotypic traits and propose functional hypotheses, despite the absence of replication.

## 3. Results

### 3.1. Antagonistic Activity of Trichoderma Strains

Dual-culture assays revealed more antagonistic activity against *M. roreri* (MR), with the ten *Trichoderma* strains showing a mean growth inhibition of ≈94% (PPI range: 89–97; Table 1). Strain C1 exhibited the highest inhibition levels against MR (97% PPI; Figure 1). Against *M. perniciosa* (MP), inhibition levels were slightly lower (mean PPI ≈ 90%, range: 86–91), with strains C2A and C4B showing consistently high activity against both pathogens (Table 1). Notably, these two strains maintained equivalent inhibition levels against both species MR and MP, suggesting broad-spectrum antagonism.

### 3.2. Chitinolytic Activity of Trichoderma spp.

Chitinase activity showed significant variation depending on substrate source. When grown on commercial shrimp-derived chitin, strain C8 exhibited exceptional activity (≈267 mg/mL NAGA), while other strains ranged from 52 to 117 mg/mL NAGA, categorized as high chitinase producers (5/10 strains), medium (2/10), and low (2/10) (Table 1). Overall, commercial crustacean chitin elicited the strongest enzymatic response across all 10 strains (mean = 99.7 mg/mL NAGA). Notably, activity levels with crustacean chitin were nearly double those observed when using chitin derived from fungal cell walls of *M. perniciosa* (mean = 48 mg/mL, range: 22–94) or *M. roreri* (mean = 41 mg/mL, range: 24–72) (Table 1). Intriguingly, some strains (C5, C10) exhibited discordant results in the semi-quantitative and quantitative assays. While producing minimal clearance zones on solid media (Figure 2), they released substantial NAGA in liquid culture (≈117 and 52 mg/mL, respectively).

### 3.3. Volatile Organic Compounds and Their Limited Association with Antagonism

The saturated fatty acid hexadecanoic acid (palmitic acid) emerged as the dominant metabolite in 10 out of 12 *Trichoderma* and *Moniliophthora* in dual- (Figure 3A,B) and solo-cultures (Figure 3C), comprising maximum peak abundance from 11.2 to 26.7% peak area. Hexadecanoic acid (HDA) was detected in 8/9 *Trichoderma* strains in confrontation with MR, and in 9/10 with MP. *T. spirale* C3B, MP, and *Trichoderma* sp. C3A showed the highest abundance of 26.7%, 18.41, and 14.6%, respectively (Table 1). Interestingly, HDA was by far more abundant in MP than in MR (0.23%) peak area. The other six compounds (eicosane; hexadecanoic acid, methyl ester; heneicosane; 9-octadecenoic acid (Z)-methyl ester; phenol 2,4-bis(1,1-dimethylethyl)-; and valencene) were detected in ≥50% of the strains. On the other hand, octadecane, 1-nonadecene, diisooctyl phthalate, bis(2-ethylhexyl) phthalate, α-selinene, and ergosterol were produced by 1 or 2 strains.

GC-MS analysis of solo-culture and dual-culture (in confrontation zone) 18 volatile compounds with distinct distribution patterns across *Trichoderma–Moniliophthora* interactions were analyzed (Appendix A). Eight compounds were detected both in solo- and dual-culture with peak areas from 0.02 to 26.65% (1-nonadecene; bis(2-ethylhexyl) phthalate; ergosterol; heneicosane; hexadecanoic acid; hexadecanoic acid, methyl ester; phenol, 2,4-bis(1,1-dimethylethyl)-; and eicosane). Five VOCs were detected only in the solo-culture of *Trichoderma* and in very low abundance (0.1 to 0.62% peak area), dibutyl phthalate was detected in three strains, and acorenone, lanosterol, phenol, 2,4-bis(1-phenylethyl)-, and phenylethyl alcohol were detected only once (in one strain each). The other five compounds: 9-octadecenoic acid (Z)-, methyl ester; octadecane; valencene; diisooctyl phthalate; and α-selinene were exclusively detected in the confrontation zone, ranging from 0.03 to 2.04% (the last two only in one strain each). In *Trichoderma* spp. solo-culture, 1–7 metabolites were detected, while in *Moniliophthora* spp., 5–6 were detected (Appendix A).

Hierarchical clustering analysis revealed distinct strain-specific VOC patterns (Figure 3). Bootstrap resampling analysis demonstrated the confidence levels for major branch resolution in *Trichoderma* clusters when confronting MR (Figure 3A), MP (Figure 3B), and in solo-culture (Figure 3C). In VOC detection against MR (Figure 3A), *T. ghanense* C4B and *T. harzianum* C4A formed clearly separated clusters (bootstrap > 64%). This divergence likely reflects their contrasting volatile detected: C4B was the only strain where hexadecanoic acid was undetected, while C4A exclusively produced this compound. Strains C1, C9, and C10 showed the highest relative abundance of HDA (indicated by red hues in Figure 3A). However, when confronting MP (Figure 3B), only C9 maintained similarly elevated hexadecanoic acid levels. Cluster resolution was notably weaker in monoculture (Figure 3C), with bootstrap values <50% for most upper branches, indicating limited statistical support for these groupings.

The estimated amount of hexadecanoic acid differed most notably between monocultures and dual cultures in C3B, particularly when co-cultured with MR, where it decreased by 4.2-fold (Table 1). C4B drastically inhibited metabolite production when grown against MR. In contrast, HDA was undetectable in monocultures of C10 and C2A, but its production increased by 5- to 10-fold when these strains were exposed to the pathogens (Table 1). Metabolic adaptation differences among *Trichoderma* strains were most pronounced in C10, C2A, C4B, C3A, and C1, whereas C5, C8, and C4A exhibited stable % peak area values. Additionally, MP produced 80 times more hexadecanoic acid in monoculture than MR, highlighting distinct interspecies variations.

### 3.4. Integrated Analysis for Biocontrol Potential Assessment

Multivariate analysis of antagonistic traits (PPI, chitinase activity, and HDA production) revealed functionally distinct *Trichoderma* strain groupings, with PCA biplot (Figure 4A) and hierarchical clustering (cophenetic correlation = 0.773) segregating strains into two ecologically meaningful clusters (Figure 4B). PCA of these traits revealed distinct strain groupings, with the first two components explaining 58% of total variance (Figure 4A). Hierarchical clustering corroborated this separation, yielding two major clades with a cophenetic correlation of 0.773 (Figure 4B). This division primarily reflected differences in hexadecanoic acid production (MR-HDA/MP-HDA) and chitinase activity (Chit-MR) and MP antagonism (PPI-MP), as visualized in the PCA biplot. Notably, PPI (PPI-MR) values contributed minimally to group separation.

Cluster I (blue), comprising *T. reesei* C2A, *T. harzianum* C4A, *T. ghanense* C4B, and *T. spirale* C5, most exhibited specialized enzymatic strategies against MR (C4A, C4B, and C5—ranged from 72 to 54 mg/mL), and low HDA production (0–5%) coupled with an inverse relationship between high MP inhibition and very low MP HDA production (Figure 4A). In contrast, Cluster II (red), containing *T. harzianum* C1 and most of *T. spirale* strains C3B/C8/C9/C10, prioritized volatile-mediated antagonism, evidenced by strong HDA abundance responses to MR and MP, and lower chitinase consistency against MR (24.2–39.2 mg/mL). This group included standout performers like C8, which achieved 96.69% MR inhibition, and C9, demonstrating broad-spectrum HDA induction. Neither cluster exhibited homogeneous metabolic signatures or uniformly superior biocontrol performance, suggesting strain-specific functional trade-offs between metabolite production and enzymatic antagonism.

Analysis underscored critical trade-offs, including a strong negative correlation (Pearson’s correlation r = −0.84, *p* < 0.05) between HDA abundance and chitinase activity against MR, pathogen-specific specialization, and the absence of a universally superior cluster—collectively suggesting that optimal biocontrol may depend on strategic strain combinations. Notably, the translational potential of C9 sustained HDA production, C4B and C1 transcend their cluster divisions, highlighting their dual value as both standalone agents and components of mixed formulations for integrated pathogen management.

## 4. Discussion

Our results showed that several *Trichoderma* strains exhibited notable biocontrol potential, as evidenced by consistent HDA production and, in some cases, stable NAGA levels when challenged with pathogens (MR or MP). Notably, eight out of ten strains produced HDA under pathogen pressure, while two were completely inhibited. The results of this study reveal a complex interplay between *Trichoderma* antagonistic mechanisms (mycoparasitism/chitinase/volatile) during confrontation with *Moniliophthora* pathogens, underlying mechanisms of mycoparasitism, antibiosis, and competition for nutrients and space [28,29]. The combination may be valuable for biocontrol applications [3,30].

### 4.1. Complementary Antagonistic Profiles Against Moniliophthora spp.

Despite low HDA levels, C2A and C4B strongly inhibited both pathogens (>93% PPI), with C4B showing robust mycoparasitism, sporulating over MR colonies in an independent study [21]. Chitinase activity was substrate- and strain-specific, indicating *Trichoderma* strain adaptation to different pathogen cell walls. C4A and C1 responded better to pathogen-derived chitin, almost the same as crustacean chitin (medium and high producers [15], respectively), potentially due to structural factors (e.g., crystallinity, particle size, or bioactive cell wall components) [15,23,31]. Such specificity reflects *Trichoderma*’s genetic capacity for diverse chitinase expression and the influence of fungal cell wall components [12,32]. While crustacean chitin generally induced greater activity on 8/10 strains, ref. [15] reported 55 times more NAGA was released from shrimp chitin than from *Rhizoctonia solani* cell wall chitin. While limited by single experimental runs, laboratory and field C2A/C4B assays [21] support the biological relevance of these strains. Observed strain-specific trends provide a solid basis for prioritizing candidates in further replicated research.

### 4.2. Volatile Organic Compounds as Secondary Contributors to Antagonism

HDA was the dominant VOC in 80% of dual-cultures, with notable pathogen-induced modulation. MR interactions suppressed (3/9) and stimulated (6/9) HDA across strains, suggesting metabolic reprogramming. The same trend was shown for MP (3/9, 5/9, respectively), while one strain (C2A) produced no HDA. HDA inducers showed no clear correlation to their antagonism, implying alternative inhibitory mechanisms. On the other hand, C10 produced HDA only when challenged with *Moniliopthora* pathogens, but not in solo-culture, nor in [20]. Contrasting with our VOC data, ethyl acetate extracts of C10 in [20] contained oleic acid as the dominant compound (7% peak area), though its role in antagonism was inconclusive (higher oleic acid present in less-inhibitory strains). Notably, in [20], C10’s efficacy seems to be attributed to the presence of quinones, which were detected qualitatively but not identified by GC-MS profiling, reinforcing the need for complementary detection methods (chromatographic parameters, extraction methods). HDA’s role aligns with known oxylipin functions in fungal biology [33], from antifungal to phytotoxic, highlighting its complex ecological function. Dose-dependent inhibition of ascomycetes (e.g., *Fusarium* spp.) via complete inhibition of mycelial growth and spore germination because of membrane disruption [34]. In basidiomycetes, HDA has shown contrasting roles: it promotes conidial germination in *Austropuccinia psidii* [35], while in *Rhizoctonia solani*, HDA acts as a pathogenicity factor in rice sheath blight, indicating potential phytotoxic effects [36]. While VOC profiles differed among strains (our findings are consistent with [20]), they generally featured one major compound and numerous minor ones, with detection highly dependent on extraction and GC-MS conditions [37,38,39].

### 4.3. Strain Specialization Suggests Potential for Tailored Consortia

Synergistic effects between VOCs and cell-wall-degrading enzymes (CWDEs) likely underpin the observed antagonism. *Trichoderma* strains combining multiple mechanisms—e.g., C4B (chitinase-rich) and C9 (HDA producer)—could enhance efficacy. The synergism plays a critical role in enzymes, e.g., *Trichoderma* strains co-expressing *β*-1,3-glucanase and chitinase suppressed *Rhizoctonia meloni* growth by 60%, outperforming single-enzyme transformants [40]. While individual compounds such as eicosane and dibutyl phthalate, and (9Z)-octadecenoic acid demonstrated direct antifungal effects [41,42], their combinatorial action with other metabolites could significantly enhance antagonistic potency. For instance, *T. harzianum* acetonic extract—containing hexadecanoic acid (12.98%), diisooctyl phthalate (10.67%), and harzianic acid (9.45%)—exhibited stronger anti-*Fusarium* activity than any single component alone [6]. CWDEs (e.g., chitinases, glucanases) disrupt pathogen cell walls, enabling secondary metabolites to penetrate and inhibit cellular processes more effectively [4]. While *Trichoderma* strains exhibited strong inhibitory effects against both MR and MP, the underlying drivers of this antagonism appear to be primarily non-volatile, with VOCs playing a less direct role than initially hypothesized. These findings align with recent literature emphasizing the importance of enzymatic and physical interactions in *Trichoderma*-mediated biocontrol through direct pathogen inhibition or plant defense modulation [7,8,43]. The lack of a universal high-performing strain supports using consortia tailored to pathogen profiles [44]. PCA and clustering confirmed complementary functional traits across strains, advocating for strategic strain combinations in biocontrol formulations.

### 4.4. Methodological Considerations and Future Directions

This exploratory study used single-run data per strain, limiting statistical comparisons but enabling broad characterization. While VOC and enzyme analyses focused on key metabolites, some potential antifungal compounds (e.g., quinones) may have been missed [20]. Future work should include (i) replicated trials of promising consortia (e.g., C4B + C9) [44], (ii) expanded enzymatic profiling given cell wall complexity [40], and (iii) multi-omics integration to clarify strain-specific biocontrol mechanisms.

## 5. Conclusions

This exploratory study highlights the biocontrol potential of diverse *Trichoderma* strains against *M. roreri* and *M. perniciosa*, revealing functional complementarity across isolates. Strains C2A and C4B demonstrated dual-pathogen efficacy with over 93% inhibition in vitro, while C4A and C1 exhibited outstanding chitinolytic activity, reaching NAGA levels of 72 mg/mL and 94 mg/mL against MR and MP, respectively. Additionally, strain C9 consistently produced high levels of hexadecanoic acid (>11% of total VOC peak area), suggesting a robust volatile-mediated antagonism mechanism against both pathogens.

Multivariate analysis revealed strain-specific specialization, notably an inverse relationship between chitinase activity and VOC production. This mechanistic insight supports the rational design of synergistic consortia—for example, combining enzymatically active strains (C4B) with VOC producers (C9)—to enhance pathogen suppression across multiple infection stages. To consolidate these findings, further research is needed, including full biological replication, expanded enzymatic and metabolomic profiling, and whole-genome sequencing to elucidate the molecular basis of antagonistic traits and improve strain selection for field application.

## Figures and Tables

**Figure 1 microorganisms-13-01499-f001:**
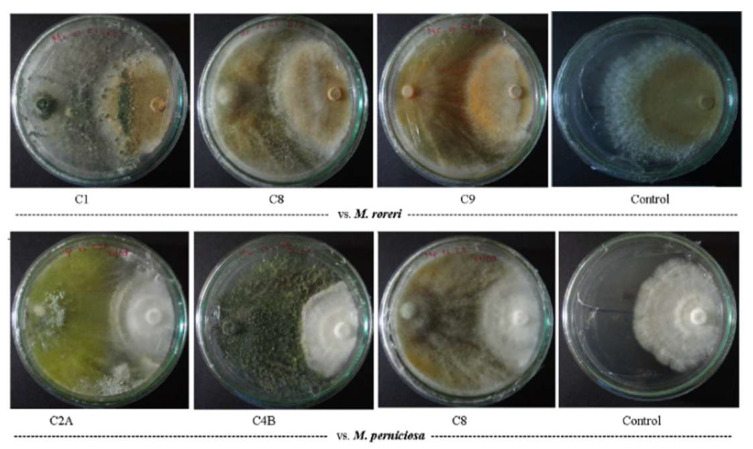
Dual-culture test of *Trichoderma* spp. against *M. roreri* (**above**) and *M. perniciosa* (**below**) on PDA plates. Pathogens inoculated on the right side.

**Figure 2 microorganisms-13-01499-f002:**
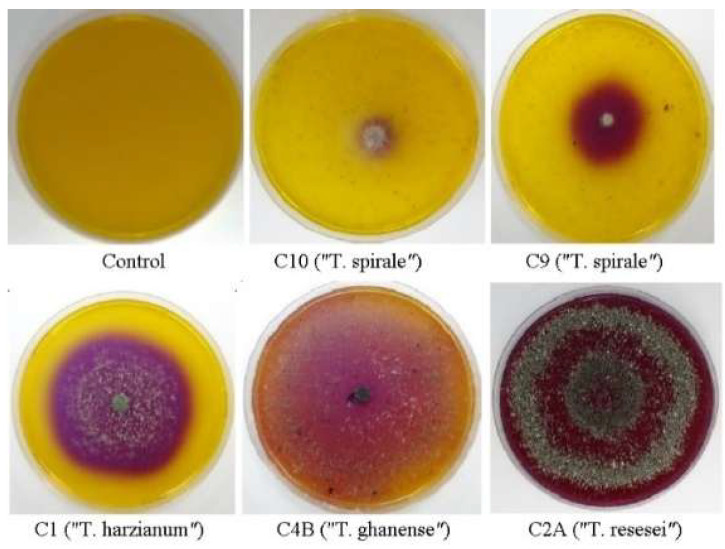
Selected *Trichoderma* chitinase-positive strains at three days of growth on basal medium supplemented with commercial chitin from shrimp shells and bromocresol purple dye.

**Figure 3 microorganisms-13-01499-f003:**
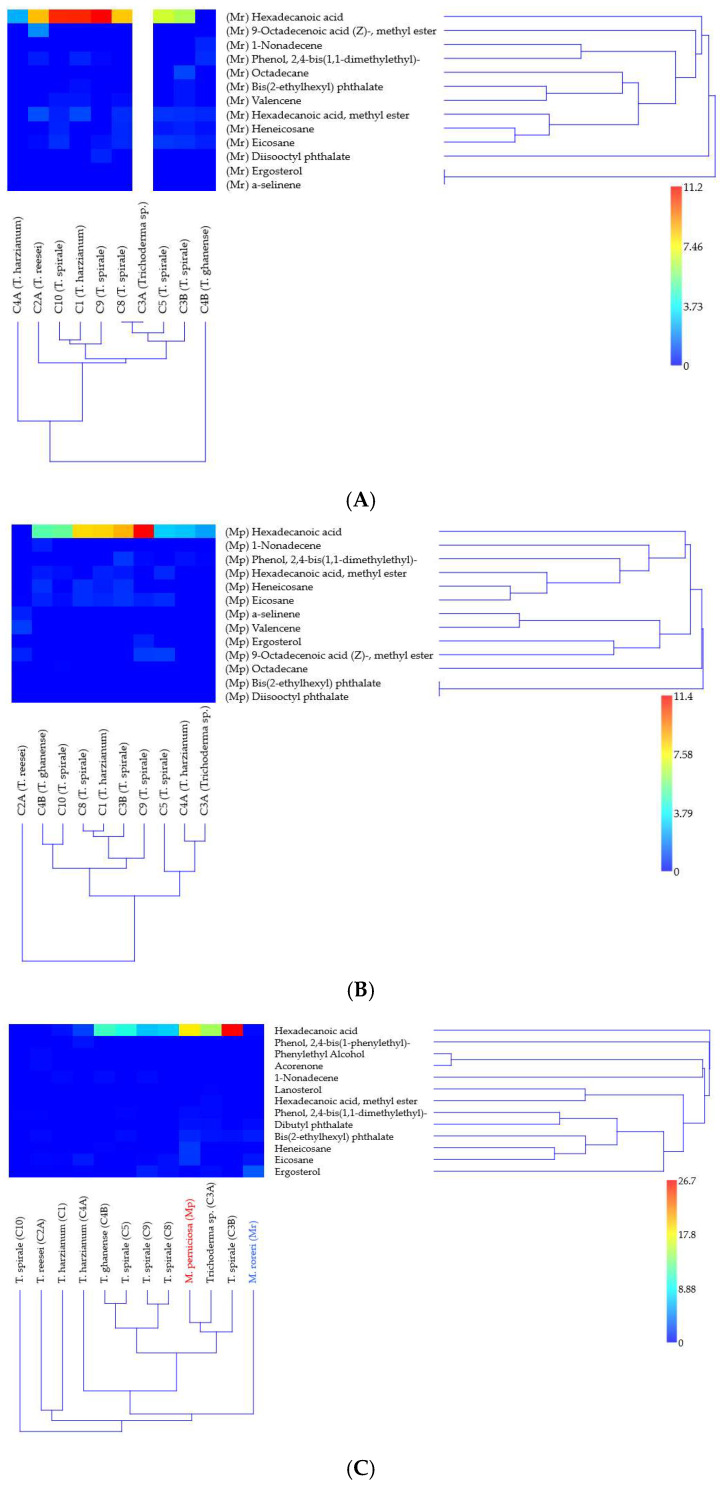
Hierarchical clustering of *Trichoderma* VOC signatures in response to *Moniliophthora* pathogens. Heatmaps show relative abundances (% total peak area) of 18 metabolites identified by GC-MS in (**A**) *M. roreri* and (**B**) *M. perniciosa* confrontations, and (**C**) monocultures. Color gradient indicates metabolite abundance (see scale). Clustering was performed using UPGMA with Bray–Curtis similarity (cophenetic correlation > 0.9); branches with bootstrap support >63% (1000 replicates) are labeled. Analyses conducted in PAST v5.2.1.

**Figure 4 microorganisms-13-01499-f004:**
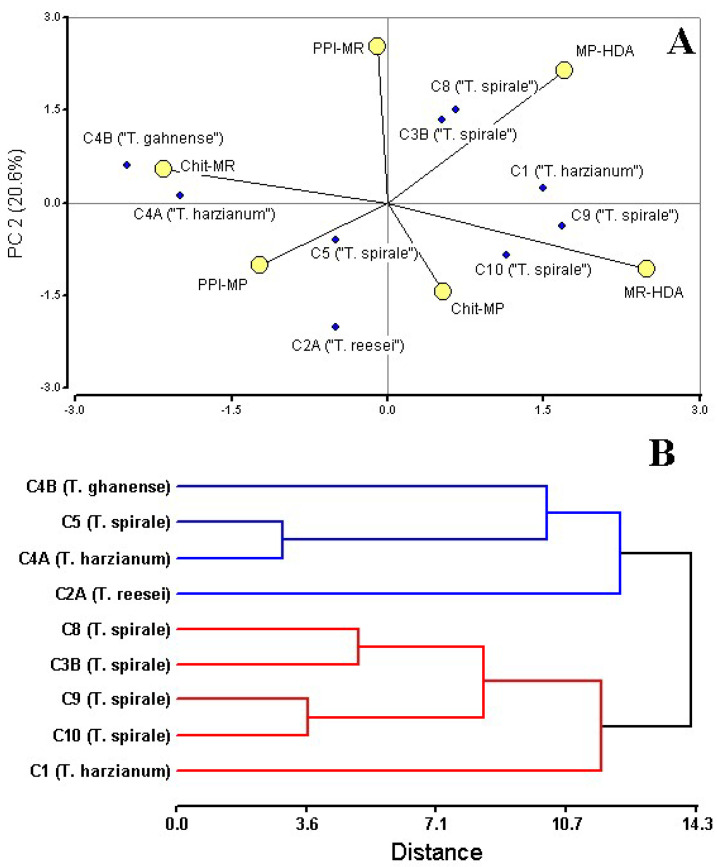
Multivariate analysis of *Trichoderma* biocontrol traits against *Moniliophthora* pathogens. (**A**) Principal Component Analysis (PCA) biplot of strain performance metrics: mycelial growth inhibition (PPI-MR, PPI-MP), chitinase activity (Chit-MR, Chit-MP), and hexadecanoic acid (MR-HAD, MP-HDA). (**B**) Hierarchical clustering dendrogram (UPGMA, Euclidean distance; cophenetic correlation = 0.773) showing strain groupings based on antagonistic profiles. Strain C3A (*Trichoderma* sp.) was not included in the analyses since VOCs were not obtained from the dual culture with MR. Analysis conducted in InfoStat ver. 2020.

**Table 1 microorganisms-13-01499-t001:** Antagonistic activity of *Trichoderma* strains against *Moniliophthora* pathogens in dual- and solo-cultures. Highlighted strains show consistent performance across both pathogens.

Strain (Species) ^1^	vs. *M. roreri*		vs. *M. perniciosa*		Solo-Culture	vs. *M. roreri*	vs. *M. perniciosa*
	% Pathogen Inhibition ^2^	Chitinase Activity ^4^ (mg/mL NAGA)	% Pathogen Inhibition	Chitinase Activity (NAGA)	Hexadecanoic Acid (% of Peak Area)		
MR (*M. roreri*)	-- ^3^	--	--	--	0.23	--	--
MP *(M. perniciosa*)	--	--	--	--	18.41	--	--
C3B (“*T. spirale*”)	94.68	39.2	87.5	31.7	26.65	6.3	8.73
C3A (*Trichoderma* sp.)	89.32	24.2	87.56	26.7	14.62	--	2.39
C4B (“*T. gahnense*”)	95.43	66.7	93.53	56.7	11.02	0	5.07
C5 (“*T. spirale*”)	92.6	54.2	88.13	49.2	9.98	6.75	3.06
C8 (“*T. spirale*”)	96.69	24.2	91	21.7	7.17	8.27	8.14
C9 (“*T. spirale*”)	91.8	34.2	89.79	54.2	6.81	11.19	11.37
C4A (“*T. harzianum*”)	92.99	71.7	89.09	34.2	2.22	2.55	2.9
C1 (“*T. harzianum*”)	95.87	39.2	88	94.2	0.55	10.67	8.23
C10 (“*T. spirale*”)	92.59	24.2	89.39	46.7	0	10.65	5.3
C2A (“*T. reesei*”)	93.04	29.2	93.5	61.7	0	8.42	0
Average	93.50	40.7	89.75	47.7			

^1^ The *Trichoderma* strains were identified to the genus level. Species designations are provisional, based solely on >99% ITS sequence similarity to reference species, and should not be interpreted as definitive taxonomic assignments. ^2^ Abbreviated in the text as PPI. PPI of *Moniliophthora* spp. colony growth by *Trichoderma* strains after 7 days of co-culture in PDA. ^3^ Dashes indicate no data. ^4^ Amount of released NAGA (N-acetyl-*β*-D-glucosamine) in colloidal chitin supplemented media by individual isolates of *Trichoderma* spp.: Low (NAGA conc. 30–60 mg/mL), Medium (NAGA conc. 61–80 mg/mL), and High (NAGA conc. > 81 mg/mL) chitinase producers [15].

## Data Availability

The raw data supporting the conclusions of this article will be made available by the authors on request.

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
