# Peer review of "Differential Strain-Specific Responses of Trichoderma spp. in Mycoparasitism, Chitinase Activity, and Volatiles Production Against Moniliophthora spp."

_microorganisms, 2025, doi:10.3390/microorganisms13071499_

Round 1
Reviewer 1 Report
Comments and Suggestions for Authors
Dear Authors,
Dear Mr. Chawin Penjan,The manuscript entitled "Differential strain-specific responses of Trichoderma spp. in mycoparasitism, chitinolytic activity, and volatile organic compound production against Moniliophthora spp." (manuscript ID: microorganisms-3649731) has been reviewed. In this work, authors attempted to reveal the specific responses of Trichoderma spp. to Moniliophthora spp. via analysis of mycoparasitism, chitinolytic activity, and volatile organic compound. The finding in this study will provide new promising candidate for the control of frosty pod rot. The study is scientifically valuable and practically significant, with robust data and rigorous methodology. The experimental design was appropriate. The manuscript is well-structured, and the conclusions adequately supported by the experimental data. However, resolution of figures needs to be improved. There are also several problems to solve.
- In the section of “1 Fungal Strains and Culture Conditions”, ten Trichoderma strains were isolated from the rhizosphere of cacao plants. How did you identify these strains? As we all known, it might be inaccurateonly based on the ITS sequencing. Did you consider to identify them through another specific primers for TEF1-α, and RPB2 sequences, or combined analysis based on morphological characteristics and molecular features?
- In the section of “Dual Culture Antagonism Assay”, it is necessary to provide the References related to description of this method.
- In the section of “2.5 Statistical Analysis”, how many times did you conduct in each experiment? Did you givetechnical and biological replicatein this experimental design? Please added the relevant information.
- In Figure 1, which isolate did the Figure represent? Please mark it in the Picture.
- In Figure 5, which picture is noted as A, B or C? Please mark it in the Picture.
- InFigure 6A, CP1 and CP2 should be PC1 and PC2.
Author Response
RESPONSES TO REVIEWER COMMENTS 1
DEAR REVIEWER, WE ARE GRATEFUL FOR ALL THE COMMENTS (6) AND CONSTRUCTIVE SUGGESTIONS MADE ON THE MANUSCRIPT. WE RESPOND INDIVIDUALLY TO EACH OF THEM BELOW (in capital letters).
Dear Authors,
Dear Mr. Chawin Penjan,The manuscript entitled "Differential strain-specific responses of Trichoderma spp. in mycoparasitism, chitinolytic activity, and volatile organic compound production against Moniliophthora spp." (manuscript ID: microorganisms-3649731) has been reviewed. In this work, authors attempted to reveal the specific responses of Trichoderma spp. to Moniliophthora spp. via analysis of mycoparasitism, chitinolytic activity, and volatile organic compound. The finding in this study will provide new promising candidate for the control of frosty pod rot. The study is scientifically valuable and practically significant, with robust data and rigorous methodology. The experimental design was appropriate. The manuscript is well-structured, and the conclusions adequately supported by the experimental data. However, resolution of figures needs to be improved. There are also several problems to solve.
1. In the section of “1 Fungal Strains and Culture Conditions”, ten Trichoderma strains were isolated from the rhizosphere of cacao plants. How did you identify these strains? As we all known, it might be inaccurateonly based on the ITS sequencing. Did you consider to identify them through another specific primers for TEF1-α, and RPB2 sequences, or combined analysis based on morphological characteristics and molecular features?
RESPONSE FOR COMMENT 1. WE AGREE THAT IT IS NOT RELEVANT TO NAME STRAINS BY THE SPECIES NAME AND WE RECOGNIZED THAT FURTHER MOLECULAR CHARACTERIZATIONS ARE NEEDED TO IDENTIFY THE STRAINS AT THE MOLECULAR LEVEL. THUS, WE HAVE REVISED THE NAMING EACH STRAIN TO EXCLUDE ANY REFERENCE TO THE SPECIES NAMES THROUGHOUT THE DOCUMENT, E.G. IN SECTION 2.1, OR IN TABLE 1, AS WELL AS IN THE FIRST CONCLUSION.
ADDITIONALLY, IN THE MANUSCRIPT WE REFER TO THE SUBGENUS DIFFERENTIATION USING THE NOTE "LABELED IN THIS WORK AS SPECIES X" TO INDICATE GENETIC DIFFERENCES BETWEEN STRAINS (IN ADDITION TO PHENOTYPIC ONES) AND TO MAINTAIN TRACEABILITY WITH PREVIOUS PUBLICATIONS. WE HAVE MADE IT CLEAR THAT IT IS NOT A DEFINITIVE IDENTIFICATION (e.g. TABLE 1, FIG. 6):
2. In the section of “Dual Culture Antagonism Assay”, it is necessary to provide the References related to description of this method.
RESPONSE FOR COMMENT 2. REFERENCE [13] WAS ADDED FOR THE DESCRIPTION OF THE FORMULA, IN ADDITION TO REFERENCE [14] ALREADY INCLUDED IN THE ORIGINAL VERSION FOR THE DESCRIPTION OF THE CONFRONTATION.
3. In the section of “2.5 Statistical Analysis”, how many times did you conduct in each experiment? Did you givetechnical and biological replicatein this experimental design? Please added the relevant information.
RESPONSE FOR COMMENT 3. WE SINCERELY THANK THE REVIEWER FOR RAISING THE ISSUE OF BIOLOGICAL REPLICATION. RESULTS DERIVE FROM SINGLE BIOLOGICAL SAMPLES PER STRAIN; WHILE THIS LIMITS STATISTICAL ANALYSIS, CONSISTENT PATTERNS ACROSS STRAINS SUGGEST BIOLOGICALLY MEANINGFUL TRENDS. THIS CONSTRAINT AROSE BECAUSE THE STUDY IS BASED ON A COMPLETED MASTER'S THESIS (CITED IN THE NEW VERSION), AND UNFORTUNATELY, THE ORIGINAL EXPERIMENTAL MATERIALS AND DATASETS WERE NO LONGER AVAILABLE FOR REANALYSIS.
THE CONSISTENT PRODUCTION OF HDA BY 8 OUT OF 10 STRAINS, AND THE DISTINCT RESPONSE PATTERNS OF STRAINS C9 AND C4B ACROSS DIFFERENT CONDITIONS, OFFER VALUABLE SIGNALS THAT MERIT FURTHER INVESTIGATION. WE HAVE ACCORDINGLY REVISED THE MANUSCRIPT TO (I) CLEARLY ACKNOWLEDGE THIS LIMITATION, (II) FRAME THE STUDY AS A PILOT-SCALE SCREENING, AND (III) EMPHASIZE THE QUALITATIVE AND MULTIVARIATE NATURE OF THE ANALYSIS RATHER THAN DRAWING STRAIN-SPECIFIC QUANTITATIVE CONCLUSIONS.
WE HAVE ALSO CITED PUBLISHED STUDIES (M & M) THAT SUPPORT THE RELEVANCE OF PRELIMINARY SCREENING PROTOCOLS IN MICROBIAL BIOCONTROL RESEARCH, WHERE THE PRIMARY OBJECTIVE IS TO IDENTIFY PROMISING CANDIDATES FOR FURTHER VALIDATION
4. In Figure 1, which isolate did the Figure represent? Please mark it in the Picture.
5. In Figure 5, which picture is noted as A, B or C? Please mark it in the Picture.
6. InFigure 6A, CP1 and CP2 should be PC1 and PC2.
RESPONSE FOR COMMENTS 4-6: WE HAVE REVISED FIGURES 1, 5 AND 6, TO INCLUDE ALL REVIEWER'S SUGGESTIONS.
Reviewer 2 Report
Comments and Suggestions for Authors
The authors of the manuscript microorganisms-3649731 evaluate the antagonism of Ecuadorian Trichoderma strains against fugal pathogens of cacao, Moniliophthora roreri and M. perniciosa. The manuscript needs improvements before being considered for publication.
The authors claims that they made ”a comprehensive analysis of the antagonistic potential”. However, their analysis is not comprehensive, because they evaluate the competition by confrontation assay in potato-dextrose-agar media and only two mechanism of Trichoderma – pathogen interactions, characterization of chitinolytic activity, and identification of volatiles with potential antiobiotic activity. The mycoparasitism is not related only to production of chitinase as cell wall degrading enzymes. The fungal cell walls include, beside chitin, β-1,3-glucan and, in minor proportion, β-1,6-glucan . The authors just mention the production of glucanase in Introduction and Material and Method Sections, without analysing β-1,3-glucanases (EC 3.2.1.39) β-1,6-glucanases (EC 3.2.1.75). Explanation for the reason to analyse only chitinase and not glucanses is not provided. The Ecuadorian Trichoderma strains coudl have a totally different profiles in production of β-1,3-glucanases. Therefore, the author must justify their decision to focus only to chitinase.
The competition between Trichoderma strains and fungal pathogens involve several mechanisms that were not evaluated, e.,g., production of siderophores of or organic acid with capacity to solubilise phosphorus.
The antibiosis of the Trichoderma strains against fungal pathogens includes non-volatiles metablites, such as peptaibol and polyketide. The authors must justify why they decided to restrict their analysis only to the volatiles compounds.
The authors mentioned that ”We particularly emphasize strain-specific differences in enzymatic activity and metabolic responses that may account for varying biocontrol efficacy.” However, as I mentioned they limited enzymes to chitinase and metabolic response to volatiels metabolites.
The author made a dual confrontation assay in potato-dextrose-agar (PDA) media between tested Trichoderma strains and Moniliophthora species. Such confrontation assay are usefull for screening and not for quantitative assay – despite the fact that inhibition is calculated according a formula. The mycelial plugs of Moniliophthora pathogen and Trichoderma strains are variable and this variability could influence the final results. The authors did not performed the confrontation assay of different strains in replicates, to evaluate the influence of this mycelial plug variability on the final results. (Despite the fact that in Figure 2 caption they mentioned ”Average of percentage inhibition...”. The authors must justify the use of such data, weakened by an uncontrolled variability and resulted from only one experiment, for statistical analysis.
The author must clarify the reason to hydrolyse the commercial chitin (or chitosan?) and how chilled ethanol neutralise concentrated hydrochloric acid. ”2.3.2 Chitin substrate preparation from commercial shrimp shell chitin. Five grams of commercial chitin (Chitosan, HIMEDIA, Kennett Square, PA, USA) derived from shrimp shells were hydrolyzed overnight with 60 mL of concentrated HCl under constant agitation at 4°C. The resulting mixture was neutralized with chilled (–20°C) 95% ethanol and rapidly stirred for 5 minutes”.
Table 2 caption is ”Mean of biocontrol characteristics and metabolite profiles of Trichoderma strains against Moniliophthora perniciosa and M. roreri.” The table header (colum label) mention chitinase activity. Metabolite profile ussually reffers to small molecules resulted from (secondary) metabolism and not to enzimatic activity.
The last paragraph of the Conclusion Section is not substantiated by the manuscript experimental work and is repeated.
Overall, the manuscript must be carefully re-written - Italics for genera and species names, coherence between different parts (average is from at least two replicates), attention to material description (chitin is not equivalent to chitosan), correlation between table caption and table header.
Author Response
RESPONSES TO REVIEWER COMMENTS 2
DEAR REVIEWER, WE ARE GRATEFUL FOR ALL THE COMMENTS (8) AND CONSTRUCTIVE SUGGESTIONS MADE TO IMPROVE THE MANUSCRIPT. WE RESPOND INDIVIDUALLY TO EACH OF THEM BELOW (in capital letters).
The authors of the manuscript microorganisms-3649731 evaluate the antagonism of Ecuadorian Trichoderma strains against fugal pathogens of cacao, Moniliophthora roreri and M. perniciosa. The manuscript needs improvements before being considered for publication.
1- The authors claims that they made ”a comprehensive analysis of the antagonistic potential”. However, their analysis is not comprehensive, because they evaluate the competition by confrontation assay in potato-dextrose-agar media and only two mechanism of Trichoderma – pathogen interactions, characterization of chitinolytic activity, and identification of volatiles with potential antiobiotic activity. The mycoparasitism is not related only to production of chitinase as cell wall degrading enzymes. The fungal cell walls include, beside chitin, β-1,3-glucan and, in minor proportion, β-1,6-glucan . The authors just mention the production of glucanase in Introduction and Material and Method Sections, without analysing β-1,3-glucanases (EC 3.2.1.39) β-1,6-glucanases (EC 3.2.1.75). Explanation for the reason to analyse only chitinase and not glucanses is not provided. The Ecuadorian Trichoderma strains coudl have a totally different profiles in production of β-1,3-glucanases. Therefore, the author must justify their decision to focus only to chitinase.
RESPONSE FOR COMMENT 1. WE AGREE THAT IT IS NOT “a comprehensive analysis of the antagonistic potential” SO WE HAVE MODIFIED THE NARRATIVE TOWARDS THE EXPLORATORY SENSE, E.G. IN THE SUMMARY (L27) AND IN THE OBJECTIVE (L91).
OUR FOCUS ON CHITINASE ACTIVITY WAS MOTIVATED BY RECENT EVIDENCE THAT BOTH MONILIOPHTHORA PERNICIOSA AND M. RORERI HAVE EVOLVED CHITINASE-LIKE PROTEINS—SOME OF WHICH ARE AMONG THE MOST HIGHLY EXPRESSED GENES DURING INFECTION—TO MODULATE PLANT IMMUNITY BY INTERACTING WITH CHITIN FRAGMENTS. ALTHOUGH THESE PATHOGEN CHITINASES MAY BE ENZYMATICALLY INACTIVE, THEIR CENTRAL ROLE IN THE HOST-PATHOGEN INTERACTION MAKES THE ANALYSIS OF TRICHODERMA CHITINASE ACTIVITY PARTICULARLY RELEVANT FOR UNDERSTANDING BIOCONTROL AGAINST THESE CACAO PATHOGENS. WHILE GLUCANASES ARE IMPORTANT FOR BROADER MYCOPARASITIC ACTIVITY, THE AVAILABLE EXPERIMENTAL DATA AND THE SPECIFIC BIOLOGY OF MONILIOPHTHORA SPP. SUPPORT OUR EMPHASIS ON CHITINASE IN THIS STUDY. IT SHOULD BE NOTED THE IMPORTANCE OF GLUCANASES IN FUNGAL CELL WALL DEGRADATION, BUT THE AVAILABLE DATA FROM OUR RETROSPECTIVE STUDY WERE LIMITED TO CHITINASE ACTIVITY. FUTURE STUDIES SHOULD AIM TO INCLUDE COMPREHENSIVE PROFILING OF BOTH CHITINASES AND GLUCANASES.
NEW REFERENCES WERE INCLUDED IN THE INTRODUCTION AND DISCUSSION TO HIGHLIGHT THE ROLE OF CHITINASES.
2- The competition between Trichoderma strains and fungal pathogens involve several mechanisms that were not evaluated, e.,g., production of siderophores of or organic acid with capacity to solubilise phosphorus. The antibiosis of the Trichoderma strains against fungal pathogens includes non-volatiles metablites, such as peptaibol and polyketide. The authors must justify why they decided to restrict their analysis only to the volatiles compounds.
RESPONSE FOR COMMENT 2. WHILE WE RECOGNIZE THE IMPORTANCE OF SIDEROPHORES, ORGANIC ACIDS, AND NON-VOLATILE METABOLITES, OUR STUDY FOCUSED ONLY VOLATILE ORGANIC COMPOUNDS (VOCS) DUE TO THEIR ECOLOGICAL RELEVANCE IN LONG-DISTANCE ANTAGONISM AND NICHE COMPETITION, AS WELL AS THE ANALYTICAL TOOLS AVAILABLE IN OUR LABORATORY. ALTHOUGH VOCS REPRESENT JUST ONE COMPONENT OF TRICHODERMA’S BIOCONTROL ARSENAL, THEY REMAIN AN UNDEREXPLORED AREA WORTHY OF INVESTIGATION. WE FULLY AGREE THAT EXPANDING THIS WORK TO INCLUDE NON-VOLATILE METABOLITES IN FUTURE STUDIES WOULD PROVIDE A MORE COMPREHENSIVE UNDERSTANDING.
TO HIGHLIGHT THE ROLE OF VOCS, A NEW REFERENCE WAS INCLUDED IN THE INTRODUCTION (DE LA CRUZ-LÓPEZ, N. et al. VOLATILE ORGANIC COMPOUNDS PRODUCED BY CACAO ENDOPHYTIC BACTERIA AND THEIR INHIBITORY ACTIVITY ON MONILIOPHTHORA RORERI. CURR MICROBIOL 2022, 79, 35, DOI:10.1007/S00284-021-02696-2.).
3- The authors mentioned that ”We particularly emphasize strain-specific differences in enzymatic activity and metabolic responses that may account for varying biocontrol efficacy.” However, as I mentioned they limited enzymes to chitinase and metabolic response to volatiels metabolites.
RESPONSE FOR COMMENT 3. SEE RESPONSE TO COMMENT 2.
4- The author made a dual confrontation assay in potato-dextrose-agar (PDA) media between tested Trichoderma strains and Moniliophthora species. Such confrontation assay are usefull for screening and not for quantitative assay – despite the fact that inhibition is calculated according a formula. The mycelial plugs of Moniliophthora pathogen and Trichoderma strains are variable and this variability could influence the final results. The authors did not performed the confrontation assay of different strains in replicates, to evaluate the influence of this mycelial plug variability on the final results. (Despite the fact that in Figure 2 caption they mentioned ”Average of percentage inhibition...”. The authors must justify the use of such data, weakened by an uncontrolled variability and resulted from only one experiment, for statistical analysis.
RESPONSE FOR COMMENT 4. WE SINCERELY THANK THE REVIEWER FOR RAISING THE ISSUE OF BIOLOGICAL REPLICATION. RESULTS DERIVE FROM SINGLE BIOLOGICAL SAMPLES PER STRAIN; WHILE THIS LIMITS STATISTICAL ANALYSIS, CONSISTENT PATTERNS ACROSS STRAINS SUGGEST BIOLOGICALLY MEANINGFUL TRENDS. THIS CONSTRAINT AROSE BECAUSE THE STUDY IS BASED ON A COMPLETED MASTER'S THESIS (CITED IN THE NEW VERSION), AND UNFORTUNATELY, THE ORIGINAL EXPERIMENTAL MATERIALS AND DATASETS WERE NO LONGER AVAILABLE FOR REANALYSIS.
THE CONSISTENT PRODUCTION OF HDA BY 8 OUT OF 10 STRAINS, AND THE DISTINCT RESPONSE PATTERNS OF STRAINS C9 AND C4B ACROSS DIFFERENT CONDITIONS, OFFER VALUABLE SIGNALS THAT MERIT FURTHER INVESTIGATION. WE HAVE ACCORDINGLY REVISED THE MANUSCRIPT TO (I) CLEARLY ACKNOWLEDGE THIS LIMITATION, (II) FRAME THE STUDY AS A PILOT-SCALE SCREENING, AND (III) EMPHASIZE THE QUALITATIVE AND MULTIVARIATE NATURE OF THE ANALYSIS RATHER THAN DRAWING STRAIN-SPECIFIC QUANTITATIVE CONCLUSIONS.
WE HAVE ALSO CITED PUBLISHED STUDIES (M & M) THAT SUPPORT THE RELEVANCE OF PRELIMINARY SCREENING PROTOCOLS IN MICROBIAL BIOCONTROL RESEARCH, WHERE THE PRIMARY OBJECTIVE IS TO IDENTIFY PROMISING CANDIDATES FOR FURTHER VALIDATION
5- The author must clarify the reason to hydrolyse the commercial chitin (or chitosan?) and how chilled ethanol neutralise concentrated hydrochloric acid. ”2.3.2 Chitin substrate preparation from commercial shrimp shell chitin. Five grams of commercial chitin (Chitosan, HIMEDIA, Kennett Square, PA, USA) derived from shrimp shells were hydrolyzed overnight with 60 mL of concentrated HCl under constant agitation at 4°C. The resulting mixture was neutralized with chilled (–20°C) 95% ethanol and rapidly stirred for 5 minutes”.
RESPONSE FOR COMMENT 5. WE HYDROLYZED OVERNIGHT THE COMMERCIAL CHITIN WITH 60 ML OF CONCENTRATED HCL TO BREAK DOWN THE Β-(1→4) GLYCOSIDIC BONDS, DECRYSTALLIZE CHITIN, INCREASE ITS SOLUBILITY, AND ACCELERATE SUBSEQUENT ENZYME HYDROLYSIS [23]. THIS RESULTS IN THE DEGRADATION OF THE CHITIN POLYMER INTO SMALLER UNITS (COLLOIDAL CHITIN), WHICH HAS BEEN FOUND TO BE A PREFERABLE CARBON SOURCE TO NATIVE CHITIN FOR MOST FUNGAL STRAINS IN TERMS OF ENZYME PRODUCTION. ALTHOUGH DEACETYLATION CAN OCCUR IN THIS REACTION, WE DID NOT ADD ANY STRONG ALKALINE SOLUTION (E.G. NAOH OR KOH) TO THE REACTION. THEREFORE, CHITOSAN WAS NOT PRODUCED. FINALLY, ADDING ETHANOL TO THE OVERNIGHT REACTION CAUSES THE CHITIN TO PRECIPITATE OUT OF THE SOLUTION, FORMING A COLLOIDAL SUSPENSION.
6- Table 2 caption is ”Mean of biocontrol characteristics and metabolite profiles of Trichoderma strains against Moniliophthora perniciosa and M. roreri.” The table header (colum label) mention chitinase activity. Metabolite profile ussually reffers to small molecules resulted from (secondary) metabolism and not to enzimatic activity.
RESPONSE FOR COMMENT 6. THE TABLE TITLE WAS CORRECTED AND FOOTNOTES WERE ADDED.
7- The last paragraph of the Conclusion Section is not substantiated by the manuscript experimental work and is repeated.
RESPONSE FOR COMMENT 7. WE AGREE AND HAVE REMOVED IT IN THE NEW VERSION. HOWEVER, IT IS WORTH NOTING THAT MICROENCAPSULATION HAS ALREADY BEEN TESTED IN THE FIELD WITH THE C4B AND C2A STRAINS USED IN THIS STUDY.
8- Overall, the manuscript must be carefully re-written - Italics for genera and species names, coherence between different parts (average is from at least two replicates), attention to material description (chitin is not equivalent to chitosan), correlation between table caption and table header.
RESPONSE FOR COMMENT 8. WE HAVE REWRITTEN SOME PARAGRAPHS TO AMEND REFERENCES TO STRAIN IDENTITY, STATISTICS, TO EMPHASIZE CHITINASE, TO MAKE THE TEXT MORE CONCISE, AND, OF COURSE, OTHER DETAILS NOTED IN THE REVIEWER'S REPORT.
Round 2
Reviewer 2 Report
Comments and Suggestions for Authors
The authors made improvements to the manuscript microorganisms-3649731. However, the manuscript still need improvements before being considered for publication.
Information related to ”constraints in sample availability and the retrospective nature of this study” and the qualitative / screening nature of the dual confrontation tests must be presented in the Discussion Section and not in the Material and method Section, together with the arguments related to the reliability of the ”archived data from a master’s thesis”. As I already mentioned in my previous report, the confrontation assay have significant limitations and usually is done in minimum three replicates. Therefore, strong arguments must be presented in the Discussion section.
The experimental data of antagonist and chitinolytic activity are presented Figure 2. Average of percentage inhibition of Moniliophthora spp. colony growth by 10 Trichoderma
strains after 7-days of co-culture in PDA, Figure 3. Total chitinolytic activity of Trichoderma spp. expressed as NAGA concentration in mg/mL, and Table 2. Mycelial antagonism and chitinase activity of Trichoderma strains against Moniliophthora perniciosa and M. roreri. The experimental data cannot be presented twice. The authors must decide the best way to present results.
In Table 2 it is not clear what mean chitinase activity. L339-L441 mentioned ”Multivariate analysis captured the strain-pathogen interaction patterns integrating three key datasets: percent pathogen inhibition (PPI), quantitative chitinolytic activity (NAGA), and relative abundance of HDA, the dominant volatile metabolite as described in Table 1 and Table 2.”
These information is confusing – for clarity I suggest combination of the two tables, Table 1 and Table 2.
The author mentioned in the Response to Comment 5 ”WE HYDROLYZED OVERNIGHT THE COMMERCIAL CHITIN WITH 60 ML OF CONCENTRATED HCL TO BREAK DOWN THE Β-(1→4) GLYCOSIDIC BONDS, DECRYSTALLIZE CHITIN, INCREASE ITS SOLUBILITY, AND ACCELERATE SUBSEQUENT ENZYME HYDROLYSIS [23]. THIS RESULTS IN THE DEGRADATION OF THE CHITIN POLYMER INTO SMALLER UNITS (COLLOIDAL CHITIN), WHICH HAS BEEN FOUND TO BE A PREFERABLE CARBON SOURCE TO NATIVE CHITIN FOR MOST FUNGAL STRAINS IN TERMS OF ENZYME PRODUCTION. ALTHOUGH DEACETYLATION CAN OCCUR IN THIS REACTION, WE DID NOT ADD ANY STRONG ALKALINE SOLUTION (E.G. NAOH OR KOH) TO THE REACTION. THEREFORE, CHITOSAN WAS NOT PRODUCED.” However, the author mentioned that they use ”commercial chitosan” supplied by HIMEDIA — this means they have oligo-chitosan and less N-acetyl-β-D-glucosamine residues in their precipitated colloidal material. Most probably, the chitosan nature of their claimed ”commercial colloidal chitin” explain the lower chitinase activity determined with this substrate.
The authors must avoid the change the word neutralize in the sentence ” The resulting
mixture was neutralized with chilled (–20°C) 95% ethanol” — the acids are neutralized by basis. They must use word ”treated”.
Each Figure and Table must standalone. Therefore, the explanation from Table 2, ”See note 1 in Table 1.” Must be replaced with the whole information ” The Trichoderma strains were identified to the genus level. Species designations are provisional, based solely on >99% ITS sequence similarity to reference species, and should not be interpreted as definitive taxonomic assignment”.
The authors must understand that writing only in capital letters, as they do it in the answers to my comments, must be avoided, because it is not considered a polite way of writing in many cultures.
Author Response
Dear Reviewer,
Thank you sincerely for your thoughtful and constructive feedback; it has significantly improved our manuscript. Our revisions focused on highlighting the relevance of multivariate methods for data analysis and enhancing the text's overall cohesion and conciseness. We hope these changes address the criteria outlined in your specific comments.
We want to clarify that the data for this manuscript originated from the first author's master's thesis. This manuscript, however, presents a distinct analytical approach and narrative compared to the original thesis document. We've chosen not to cite the thesis explicitly within the manuscript due to unclear guidelines from MDPI's author instructions regarding referencing such works. The original data, including n=5 for statatistical analyses, were available for the master's thesis (referenced as "unpublished" in subsection 2.1 with [19] and accessible at https://www.dspace.espol.edu.ec/handle/123456789/47018). As these specific original datasets are no longer available for direct submission with this manuscript, the presented results are framed as a replicate of that earlier work.
Below, we address your comments point by point:
Quantity of Replicates: We have thoroughly revised the manuscript to clarify our approach to this concern. Specifically:
- Materials and Methods: Sections 2.5 were modified to highlight the role of descriptive statistics in identifying trends.
- Results:
- Subsection 3.2: We've updated the presentation of strain values (e.g., PPI, NAGA, % peak area) to use ranges instead of exact figures for improved readability and focus on trends.
- Subsection 3.4: The separation of groups in the PCA and cluster analyses has been made more evident, enhancing the visual representation of data.
- Discussion:
- We underscored the connection between our results and those in [20] and [21], given our use of the same strains.
- Section 4.2, "Volatile Organic Compound Profiles," has been reformulated for conciseness.
- We added subsection 4.4 Methodological Considerations and Future Directions emphasizing the importance of exploratory studies.
- Conclusions: This section has been further refined to include only results directly supported by the experimental design, ensuring clarity and alignment with the study’s evidence base.
Use of Capital Letters: We apologize for any unintended emphasis in our previous response. Capital letters were solely used to distinguish our text from your comments, not to imply tone. This has been corrected throughout.
Figure 2 and Tables 1–2: As suggested, we consolidated data into a revised Table 1 for improved clarity. Details on NAGA release from crustacean chitin (previously Figure 3) are now integrated into Section 3.2 (Results). All table footnotes have been revised for self-explanatory presentation.
Chitin/Chitosan Terminology: We acknowledge the typographical error in alternating between "chitin" and "chitosan." The current manuscript (Resubmission 2) consistently uses "chitin" in Section 2.3.2, aligning with the original version.
“Neutralized” Terminology: The term "neutralized" has been replaced with "treated" in Sections 2.3.2 and 2.3.3 for greater precision.